# Buffering Capacity Comparison of Tris Phosphate Carbonate and Buffered Peptone Water *Salmonella* Pre-Enrichments for Manufactured Feed and Feed Ingredients

**DOI:** 10.3390/ani13193119

**Published:** 2023-10-06

**Authors:** Cesar Escobar, Luis R. Munoz, Matthew A. Bailey, James T. Krehling, Wilmer J. Pacheco, Rüdiger Hauck, Richard J. Buhr, Kenneth S. Macklin

**Affiliations:** 1Department of Poultry Science, College of Agriculture Auburn University, Auburn, AL 36849, USA; cze0034@auburn.edu (C.E.); lrm0029@auburn.edu (L.R.M.); bailema@auburn.edu (M.A.B.); krehjt@auburn.edu (J.T.K.); wjp0010@auburn.edu (W.J.P.); ruediger.hauck@auburn.edu (R.H.); 2Department of Pathobiology, College of Veterinary Medicine, Auburn University, Auburn, AL 36849, USA; 3USDA-ARS, US National Poultry Research Center, Athens, GA 30605, USA; jeff.buhr@ars.usda.gov; 4Department of Poultry Science, College of Agriculture Mississippi State University, Starkville, MS 39762, USA

**Keywords:** pre-enrichment, buffer, BPW, TPC, *Salmonella*

## Abstract

**Simple Summary:**

This study compared two pre-enrichment mediums, tris phosphate carbonate (TPC) and buffered peptone water (BPW), for detecting *Salmonella* in animal feed. Samples (269) were collected from different feed mills and assessed for pH changes after a 24 h incubation. Different feed ingredients showed varying initial and final pH values. Meat and bone meal had higher final pH values in both TPC and BPW, while soybean and peanut meal had lower final pH values. BPW was more effective at recovering *Salmonella* and showed greater pH changes. Four *Salmonella* isolates were recovered from meat and bone meal samples, with BPW identifying more isolates. Overall, the study highlights the importance of choosing the right pre-enrichment medium for accurate *Salmonella* detection in animal feed.

**Abstract:**

Various culture-based methods to detect *Salmonella* in animal feed have been developed due to the impact of this bacterium on public and animal health. For this project, tris phosphate carbonate (TPC) and buffered peptone water (BPW) buffering capacities were compared as pre-enrichment mediums for the detection of *Salmonella* in feed ingredients. A total of 269 samples were collected from 6 feed mills and mixed with the pre-enrichments; pH was measured before and after a 24 h incubation. Differences were observed when comparing pH values by sample type; DDGS and poultry by-product meal presented lower initial pH values for TPC and BPW compared to the other samples. For both TPC and BPW, meat and bone meal presented higher final pH values, while soybean meal and peanut meal had lower final pH values. Furthermore, for BPW, post cooling, pellet loadout, and wheat middlings reported lower final pH values. Additionally, most feed ingredients presented significant differences in pH change after 24 h of incubation, except DDGS. From meat and bone meal samples, four *Salmonella* isolates were recovered and identified: three using BPW and one using TPC. TPC provided greater buffer capacity towards neutral pH compared to BPW, but BPW was more effective at recovering *Salmonella*.

## 1. Introduction

Detecting particular microorganisms from specific sources is a problem, and proper consideration must be given to the influence of environmental factors prior to detection [1]. Exposure to freezing, heating, or freeze-drying negatively impacts attempts at detection and enumeration of some microorganisms because of physiologically weakened or injured cells [2,3,4]. Most bacteria face life-threatening and hostile conditions in their natural habitats, such as oxidation, heavy metals, DNA-damaging agents, osmolarity, starvation, weak acids, and a wide range of temperatures or pH values other than those required for optimal growth [5]. Standard cultural procedures for the isolation of *Salmonella* generally include pre-enrichment of samples in a non-selective broth medium, followed by enrichment in a selective broth medium, isolation by presumptive screening, and serological confirmation of presumptive isolates [6]. A pre-enrichment medium should not only be a noninhibitor for *Salmonella* growth but also be capable of supporting the proliferation of small numbers of these bacteria, especially in higher dilutions where potential nutritional or inhibitory substances from the food being tested have been diluted [7]. For example, according to the HIMEDIA^®^ technical data sheet [8], BPW is a pre-enrichment medium intended to support the recovery of sub-lethally injured *Salmonella* before transfer to a selective medium. Pre-enrichment media contain no inhibitors, are well buffered, and offer conditions for resuscitation of cells injured by food preservation processes. According to the composition presented in the same data sheet [8], proteose peptone provides nitrogenous and carbonaceous compounds, long-chain amino acids, and vitamins (essential nutrients). Sodium chloride maintains the osmotic balance of the medium. Disodium phosphate and mono-potassium phosphate are soluble in water and provide a high buffer capacity, which prevents sudden drops in the pH of the solution.

The optimal temperature for *Salmonella* growth is 37 °C with a range of 5 to 47 °C, and the optimum pH is 6.5 to 7.0 with a range of 4 to 9 [9,10,11]. Animal feed is a potential source of *Salmonella* [12,13,14,15,16,17]. However, a low percentage of samples tested are reported as positive for *Salmonella* [14,16]. Production of animal feed in feed mills is usually in large quantities as bulk material in batchwise production; *Salmonella* cell numbers in feed are usually low and poorly distributed [18]. Maciorowski et al. [19] mentioned that it is crucial to provide injured and stressed *Salmonella* cells with an opportunity to recover and multiply in pre-enrichment and enrichment before their isolation from animal feed. However, its detection is very important since feed ingredients can be transmitters of *Salmonella* to manufactured feed and subsequently to live production; for example, poultry and products like eggs and meat would be potential sources of these bacteria [20,21].

Richardson et al. [22] observed that *Salmonella* exposed to pH ranging from 4 to 7 on xylose lysine tergitol 4 agar can lose its ability to produce hydrogen sulfide, and this effect was dependent on the pH, stress status, and serotype. It has also been reported that low pH alters the biochemical pathways of *Salmonella* [22,23]. Additionally, different researchers have reported that detection of *Salmonella* in feed during pre-enrichment or enrichment can be complicated by the level of background microflora in the sample; the higher the level of background microflora, the lower the recovery of *Salmonella* [18,24,25].

In 2015, Berrang et al. [26] tested the buffering capacity of pre-enrichment media with different concentrations of buffer components. They reported the use of 1.0% peptone water buffered with sodium chloride (NaCl), disodium phosphate (Na_2_HPO_4_), sodium phosphate (NaHPO_4_), 1 M tris pH 8, and sodium carbonate (Na_2_CO_3_) in different combinations. The combination “tris phosphate carbonate”, which was named TPC, presented the best buffering chemistry, having a pH drop at 24 h close to 6.5 in comparison to phosphate without Tris, which suffered a pH drop at 24 h close to 5.2. In another study, Richardson et al. [27] showed that TPC had the best buffering capacity, maintaining a near-neutral pH on a variety of ingredients and feed types during incubation among the five pre-enrichment media tested (lactose broth “LB”, buffered peptone water “BPW”, double-strength buffered peptone water “2 x BPW”, universal pre-enrichment broth “UPB”, and tris phosphate carbonate “TPC”). The buffer capacity of TPC was highlighted among other pre-enrichments; however, the samples were not tested for *Salmonella* by applying further steps like enrichment, plating in selective agar, an agglutination test, or biochemical confirmation.

BPW is the standard pre-enrichment used for the recovery of *Salmonella*. Based on the previous information, the aims of this study were to evaluate the buffer capacity of TPC and BPW using animal-manufactured feed and feed ingredients and to evaluate TPC as a pre-enrichment medium to recover and identify *Salmonella* serovars compared to BPW.

## 2. Materials and Methods

Samples were collected from 6 feed mills for this study (Table 1). Letters A, B, C, D, E, and F were assigned to them. Feed mills A and B belonged to integrators and produced feed for swine and poultry production, respectively. Feed mills C, D, and F were toll mills (providing custom milling, mixing, and blending services to other companies) intended for swine production, and feed mill E was a research and education mill. The samples collected (a total of 269) were ground corn, DDGS (distillers dried grains with solubles), wheat middlings, peanut meal, soybean meal, poultry by-product meal, meat and bone meal, and manufactured feed. The manufactured feed consisted of a post-mixing mash loadout and a post-cooling pellet loadout. Ground corn was collected from the ground corn silo, and the other feed ingredients were obtained from storage containers or directly from trucks/trains during unloading. Manufactured feed samples were collected during loadout from different batches and at specific intervals of time (every 2 min). Representative samples of approximately 150 g were placed into sterile Whirl-Pak^TM^ standard bags (Nasco^®^, Fort Atkinson, WI, USA) and stored in coolers or a refrigerator until they were ready to be shipped to Auburn University. Once the samples were at Auburn University, they were stored in a refrigerated cool room (4 °C).

The TPC formula (Table 2) was consulted with the authors of [26]. All solid components were added to a sterilized beaker along with 600 mL of deionized water and mixed. Then, 100 mL of 1 M Tris pH 8.0 was added along with deionized water to bring the final volume to 1 L. The pH of the TPC solution was measured and adjusted using drops of 6N hydrochloric acid (HCl) to reach a final pH of 8.0. Lastly, the TPC solution was sterilized using a Corning sterile filter system with a membrane pore size of 0.22 μm (Corning Incorporated, Corning, NY, USA). The initial pH of the BPW solution (BD Difco, Franklin Lakes, NJ, USA) was 7.0.

From each feed sample, 2.5 g of sample was added to 22.5 mL of TPC and BPW in 50 mL polypropylene centrifuge tubes, vortexed with a VWR^®^ vortex mixer for 30 s, and then incubated for 24 h at 37 °C. The initial pH (0 h) was measured with a VWR sympHony B10P benchtop pH meter (VWR Chemicals, Fountain Pkwy, OH, USA) after mixing, and the final pH (24 h) was measured after incubation. After incubation, 1 mL of the samples were transferred to 5 mL of tetrathionate brilliant green broth (HiMedia Laboratories, West Chester, PA, USA) in 15 mL polypropylene centrifuge tubes and incubated for 24 h at 37 °C. Thereafter, all the samples were streak-plated onto xylose lysine tergitol 4 agar (XLT4: Criterion^TM^, Hardy Diagnostics, Santa Maria, CA, USA) and incubated for 24 h at 37 °C for subsequent visual identification of typical black “fish-eye” *Salmonella* colonies associated with this medium. From the suspect *Salmonella* samples in the media agar, single colonies were transferred (streak plate) to another XLT4 plate and incubated. Next, after a second visual confirmation, a single colony was transferred to a *Salmonella* ChromoSelect agar (Sigma-Aldrich Corporation, St. Louis, MO, USA) and incubated for 24 h at 37 °C. With the visual confirmation in the ChromoSelect agar, a single colony from the plate was inoculated onto Triple Sugar Iron (BD Difco, Franklin Lakes, NJ, USA), Lysine Iron agar (BD Difco, Franklin Lakes, NJ, USA), and Urea agar slants (BD Difco, Franklin Lakes, NJ, USA) and incubated for 24 h at 37 °C for biochemical confirmation.

From the same samples on the ChromoSelect agar, an agglutination test for serological confirmation was performed first using polyvalent serum A-Vi for *Salmonella* spp. (BD Difco, Franklin Lakes, NJ, USA), and then, based on serology from *Salmonella* antiserum Poly Groups A, B, C, D, and E (Difco, BD) and antiserum B, C_1_, C_2_, D_1_, E, or K (BD Difco, Franklin Lakes, NJ, USA), samples were serogrouped. Next, after agglutination testing, the isolates were shipped to the National Veterinary Services Laboratory in Ames, IA, for serovar characterization.

Data were analyzed using a generalized linear mixed model, Proc Glimmix (significant *p* ≤ 0.05), means were separated by sample type using Tukey’s HSD, and pH change in time (initial–final) was analyzed using a dependent (paired) t-test for each sample type in SAS^®^ 9.4 software.

## 3. Results

### 3.1. pH Values

The average initial and final pH values of each feed type using both pre-enrichments are presented in Table 3. The initial pH of the samples with TPC was above neutral pH (~7.0), while the samples with BPW yielded mixed values. Conversely, the final pH (after 24 h of incubation) of the samples with TPC was in the range of 5.75 to 7.09, while the samples with BPW reported values below pH 6.5.

The samples tested for this study were not exposed to acidic pH environments at the laboratory, and the lowest pH values observed after incubation were 5.59 (soybean meal, using TPC) and 4.68 (peanut meal, using BPW), as shown in Table 3.

Differences were observed when comparing pH values by sample type (*p* < 0.001). For initial TPC pH, DDGS (6.98) and poultry by-product meal (7.60) presented lower pH values; moreover, for final TPC pH readings, meat and bone meal (7.77) presented the higher pH value, while peanut meal (5.75) and soybean meal (5.59) were lower compared to the other feed types. Initial BPW pH readings showed that DDGS (5.81) and poultry by-product meal (6.54) also presented lower pH values. Furthermore, for final BPW pH, meat and bone meal (6.45) presented a higher pH value; however, pellet loadout (5.05), post cooling (5.11), wheat middlings (5.15), and soybean meal (4.86) presented lower pH values. Richardson et al. [27] reported similar pH values after 24 h of incubation using TPC and BPW for meat and bone meal of 7.88 and 6.59, respectively.

### 3.2. pH Differences

The difference between initial and final mean pH values expressed in percentages using pre-enrichment media TPC and BPW is presented in Table 4. Wheat middlings, peanut meal, soybean meal, and manufactured feed, which includes pellet loadout, post cooling, post mixing, and mash loadout, presented a higher pH difference after 24 h of incubation, with a drop of at least 17% to 32% with both pre-enrichments.

Using a dependent (paired) *t*-test, highly significant differences (*p* < 0.001) were found in most of the feed ingredients but not DDGS. This means that the initial and final pH values were significantly different from each other for the mentioned feed samples. It is also important to note that DDGS samples maintained their pH after incubation, with a slight drop using TPC (6.98 to 6.91) and a slight increase using BPW (5.81 to 5.90).

### 3.3. Salmonella Detection

All the feed samples tested were negative for *Salmonella*, except for meat and bone meal. From the meat and bone meal samples, four were positive for *Salmonella* (Table 5), that is, three samples using BPW and one using TPC. The isolates were identified as *Salmonella* Oranienburg, *S.* Senftenberg, *S.* Agona, and *S.* Infantis. The initial and final pH for *S.* Oranienburg was 6.89–6.40, *S.* Senftenberg 6.89–6.45, and *S.* Agona 7.09–6.42 using BPW. Using TPC, the initial and final pH values for *S.* Infantis were in the range of 7.80–7.78.

## 4. Discussion

According to Jay [9], pH 6.5 is near the ideal pH range for *Salmonella* growth. Berrang et al. [26] reported a pH close to 6.5 after 24 h of incubation using TPC in standard broiler feed, highlighting the buffering chemistry capacity of this pre-enrichment. Richardson et al. [22] reported that the pH impact on *Salmonella* was dependent on the serotype and the stress status of the microorganism (liquid sample or dry sample). They showed that a pH of 4.85 was required to kill 50% of *S. Typhimurium* in a non-stressed or liquid state during 24 h incubation, while in a stressed state, a pH of 5.85 was low enough to kill 50% of the cells.

Richardson et al. [27] reported similar pH values after 24 h of incubation using TPC and BPW for meat and bone meal, with pH values of 7.88 and 6.59, respectively. In this study, the pH values were 7.77 and 6.45, respectively. Cox et al. [28] reported pH values of 4.3 for DDGS, 4.8 for ground corn, and 4.6 for wheat middlings after 24 h of incubation using BPW, which differs from the data presented here (5.90, 6.01, and 5.15, respectively). It is important to note that, like other feed ingredients, DDGS and wheat middlings are by-products that vary in composition since the raw material used can be from different varieties and different environmental conditions, and the techniques to obtain the mentioned by-products can be different [29,30]. Therefore, it would be challenging to compare such samples from different feed mills or regions. For example, DDGS can be obtained by two different techniques: dry grind processing and wet milling [31]. Wheat middlings are obtained by mechanical and pneumatic methods applied to separate endosperm particles from the germ and bran; as a result, several fractions in different amounts are obtained, such as screenings, bran, middlings, shorts, and red dog [32]. The pH values reported by Cox et al. [28] may differ because different sources of feed ingredients were used for their experiment.

Wheat middlings contain highly fermentable carbohydrates [33], peanut meal and soybean meal are excellent sources of protein [34,35], and finished feed is manufactured to ensure that the animal receives all the required nutrients and supplements [36]. Therefore, high bacterial activity (fermentation) was expected during incubation using non-selective pre-enrichments, resulting in lower final pH values. The common end-products of bacterial fermentation are lactic acid, formic acid, acetic acid, butyric acid, butyl alcohol, acetone, ethyl alcohol, carbon dioxide, and hydrogen, which together decrease the pH of the medium [37].

DDGS are a coproduct of ethanol-producing factories that utilize corn and wheat as raw materials that contain high levels of crude protein, oil, and fiber [38]. DDGS fiber is composed mostly of cellulose, hemicelluloses, lignin, and pectin, which are complex carbohydrates that are difficult to metabolize [39]. It is also known that DDGS are used as a fermentation feedstock, but pre-treatment methods are needed to break down the lignin impediment, which is recalcitrant to bacterial attack and can result in the release of inhibitory products for microbes [40,41]. DDGS also can contain sulfuric acid, which is used for the control of pH during ethanol fermentation and the cleaning process of bioethanol production [42]. Sulfuric acid can inhibit or even kill microorganisms, including bacteria and fungi, due to its strong oxidizing corrosiveness [43]. Perhaps the combination of the characteristics of DDGS affected the bacterial activity during incubation (degradation), resulting in almost no pH change.

However, it is unknown how severe the procedures (stressors) applied to the different feed samples before arriving at the laboratory, for example, drying of grains, milling, pelleting, rendering processes, or chemical treatments for by-products were. There is evidence that *Salmonella* serovars respond differently to sub-lethal stressors, and the surviving cells may present greater resistance to further treatments than untreated cells [44].

Several researchers state that the detection of *Salmonella* in feed during pre-enrichment or enrichment can be negatively affected by the level of background microflora [18,24,25].

It is not clear why BPW was more effective in recovering *Salmonella* for this study; however, according to these results, it can be suggested that higher pH values can interfere in the recovery of this bacteria. As mentioned before, the optimum pH range for *Salmonella* growth is 6.5–7.0 [10,11,27], and on average, all the meat and bone samples using TPC had initial and final pH values of 7.85 and 7.77, respectively.

The four serovars identified are common in poultry production and can cause foodborne illness in humans but do not cause sickness in poultry. However, poultry can act as reservoirs of these bacteria [45,46,47]. Meat and bone meal samples belonged to the same batch; however, the samples were taken from different locations on the same container. Meat and bone meal is a by-product obtained from the rendering industry after a process of cooking ground mammal carcasses, removing fat, and drying [48]. The rendering process includes a heat step that kills microorganisms; therefore, the survival of *Salmonella* is most unlikely after processing [49]. Rendered products and finished feed are most likely to be contaminated with *Salmonella* from rodents and fomites within the processing plants and feed mills [50] or during storage or transportation to other locations. *Salmonella* can survive for long periods in dried products, such as animal feed [51]. A survey covering 1 year (2010) tested a variety of render and blender operations across the United States and Canada, showing that the contamination for *Salmonella* in the rendered animal meals produced in North America was 8% [49]. Therefore, special attention must be given to avoid recontamination during transportation, storage, handling, and distribution of this feed ingredient as well as other feed ingredients. It is essential to apply good hygiene practices in the feed mill environment to reduce the prevalence of *Salmonella* [52].

## 5. Conclusions

In conclusion, TPC provided greater buffer capacity for neutral pH compared to BPW. Manufactured feed samples (post mixing, post cooling, pellet loadout, and mash loadout), wheat middlings, peanut meal, and soybean meal experienced a pH drop after 24 h of incubation of at least 17 to 32%. Three *Salmonella* isolates were recovered from meat and bone meal samples using pre-enrichment BPW, compared to one in TPC. Further research of feed inoculated with a known *Salmonella* strain and concentration and assayed with pre-enrichment TPC and BPW is necessary to determine their efficacy in recovering *Salmonella.*

## Figures and Tables

**Table 1 animals-13-03119-t001:** Types of feed samples collected in different feed mills.

Feed Mill	Type of Feed Mill	State	Type of Feed Sample	No. of Samples
A	Pigs, Integrator	OK	Ground corn	10
			Wheat middlings	10
			Post mixing	10
			Post cooling	10
			Pellet loadout	10
B	Broilers, Integrator	MS	Ground corn	8
C	Pigs, Toll Mill	IA	Ground corn	10
			DDGS ^1^	10
			Post mixing	10
			Mash loadout	8
D	Pigs, Toll Mill	IA	Ground corn	10
			DDGS ^1^	10
			Post mixing	10
			Mash loadout	10
E	Research and Education (R&E)	AL	Ground corn	14
			DDGS ^1^	14
			Poultry by-product meal	14
			Meat and bone meal	7
			Peanut meal	7
			Post mixing	14
			Post cooling	14
			Pellet loadout	14
F	Pigs, Toll Mill	IL	Ground corn	7
			DDGS ^1^	7
			Soybean meal	7
			Mash loadout	14

^1^ Distiller’s dried grains with solubles (DDGS).

**Table 2 animals-13-03119-t002:** Tris phosphate carbonate (TPC) formula.

TPC Formula	Brand	Amount (1 L)
Peptone	BD Bacto, Franklin Lakes, NJ, USA	10 g (1%)
NaCl (sodium chloride)	VWR Chemicals, Fountain Pkwy, OH, USA	5 g (0.085 M)
Na_2_HPO_4_ (disodium phosphate)	VWR Chemicals, Fountain Pkwy, OH, USA	3 g (25 mM)
NaHPO_4_ (sodium phosphate)	Fisher Scientific, Fair Lawn, NJ, USA	1.5 g (11 mM)
Na_2_CO_3_ (sodium carbonate)	Fisher Scientific, Fair Lawn, NJ, USA	4.2 g (50 mM)
1 M Tris, pH 8.0	VWR Chemicals, Fountain Pkwy, OH, USA	100 mL (100 mM)
H_2_O	-	ad 1000 mL

Adapted from [26].

**Table 3 animals-13-03119-t003:** Mean separation of pH values by feed type using pre-enrichment media TPC and BPW.

Feed Type	No. of Samples	TPC (pH)				BPW (pH)			
		Initial	S.E. ^2^	Final	S.E. ^2^	Initial	S.E. ^2^	Final	S.E. ^2^
Ground corn	59	8.03 ^a^	0.02	7.14 ^b^	0.04	7.05 ^a^	0.02	6.01 ^ab^	0.05
DDGS ^1^	41	6.98 ^d^	0.02	6.91 ^cd^	0.05	5.81^d^	0.03	5.90 ^bc^	0.06
Poultry by-product meal	14	7.60 ^c^	0.04	7.14 ^bc^	0.08	6.54 ^c^	0.05	6.22 ^ab^	0.1
Wheat middlings	10	7.94 ^ab^	0.05	6.57 ^ef^	0.09	7.09 ^a^	0.06	5.15 ^d^	0.12
Meat and bone meal	7	7.85 ^ab^	0.06	7.77 ^a^	0.11	6.96 ^ab^	0.07	6.45 ^a^	0.15
Peanut meal	7	8.04 ^a^	0.06	5.75 ^g^	0.11	6.92 ^ab^	0.07	4.68 ^d^	0.15
Soybean meal	7	7.93 ^ab^	0.06	5.59 ^g^	0.11	6.99 ^ab^	0.07	4.86 ^d^	0.15
Post mixing	44	7.99 ^a^	0.02	6.54 ^f^	0.04	6.91 ^ab^	0.03	5.69 ^c^	0.06
Mash loadout	32	7.78 ^b^	0.03	6.80 ^de^	0.05	6.82 ^b^	0.03	5.93 ^abc^	0.07
Post cooling	24	8.04 ^a^	0.03	6.39 ^f^	0.06	6.92 ^ab^	0.04	5.11 ^d^	0.08
Pellet loadout	24	8.02 ^a^	0.03	6.23 ^f^	0.06	6.87 ^ab^	0.04	5.05 ^d^	0.08
***p*-value**		**<0.0001**		**<0.0001**		**<0.0001**		**<0.0001**	

^a–g^ Values in columns with different superscripts are significantly different from each other (*p* < 0.05). ^1^ DDGS (distiller’s dried grains with solubles). ^2^ Standard error.

**Table 4 animals-13-03119-t004:** Dependent *t*-test for the pH difference before and after incubation (24 h) of feed ingredients and finished feed using pre-enrichment media TPC and BPW.

Type of Sample	No. of Samples	TPC (pH)					BPW (pH)				
		Initial	Final	I − F (%) ^1^	S.E. ^2^	*p*-Value	Initial	Final	I − F (%) ^1^	S.E. ^2^	*p*-Value
Ground corn	59	8.03 *	7.14 *	11.11%	0.03	<0.0001	7.05 *	6.01 *	14.64%	0.07	<0.0001
DDGS ^3^	41	6.98	6.91	1.01%	0.04	0.076	5.81	5.90	−1.55%	0.06	0.164
Poultry by-product meal	14	7.60 *	7.14 *	6.08%	0.06	<0.0001	6.54 *	6.22 *	4.89%	0.02	<0.0001
Wheat middlings	10	7.94 *	6.57 *	17.25%	0.06	<0.0001	7.09 *	5.15 *	27.40%	0.06	<0.0001
Meat and bone meal	7	7.85 *	7.77 *	1.06%	0.02	0.004	6.96 *	6.45 *	7.33%	0.03	<0.0001
Peanut meal	7	8.04 *	5.75 *	28.46%	0.13	<0.0001	6.92 *	4.68 *	32.36%	0.16	<0.0001
Soybean meal	7	7.93 *	5.59 *	29.48%	0.13	<0.0001	6.99 *	4.86 *	30.46%	0.09	<0.0001
Post mixing	44	7.99 *	6.54 *	18.20%	0.03	<0.0001	6.91 *	5.69 *	17.68%	0.06	<0.0001
Mash loadout	32	7.78 *	6.80 *	12.55%	0.11	<0.0001	6.82 *	5.93 *	13.03%	0.09	<0.0001
Post cooling	24	8.04 *	6.39 *	20.52%	0.05	<0.0001	6.92 *	5.11 *	26.18%	0.03	<0.0001
Pellet loadout	24	8.02 *	6.23 *	22.22%	0.06	<0.0001	6.87 *	5.05 *	26.49%	0.05	<0.0001

* The means of the two sets of pH values with an asterisk are significantly different from each other (*p* < 0.05). ^1^ I − F (%): Difference (initial minus final) of initial and final mean values (pH) expressed as a percentage of the initial value. ^2^ Standard error. ^3^ Distiller’s dried grains with solubles (DDGS).

**Table 5 animals-13-03119-t005:** *Salmonella* isolates recovered using pre-enrichment TPC and BPW.

Sample	P.E. ^1^	pH		Agglutination Test		Serotype
		Initial	Final	Poly	Group	
Meat and bone meal	BPW	6.89	6.4	A	C1	Oranienburg
	BPW	6.89	6.45	B	E	Senftenberg
	BPW	7.09	6.42	A	B	Agona
Meat and bone meal	TPC	7.8	7.78	A	C1	Infantis

^1^ Pre-enrichment.

## Data Availability

The data presented in this study are available on request from the corresponding author.

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
