# Peer review of "Buffering Capacity Comparison of Tris Phosphate Carbonate and Buffered Peptone Water Salmonella Pre-Enrichments for Manufactured Feed and Feed Ingredients"

_animals, 2023, doi:10.3390/ani13193119_

Round 1

Reviewer 1 Report

In this manuscript, two non-selective enrichment methods for the detection of Salmonella in feed are compared. Salmonella entry into livestock farms through contaminated feed has great impact, therefore sensitive detection methods are very important and should also be continuously improved.

The authors compared the standard pre-enrichment broth BPW (buffered peptone water) and Tris phosphate carbonate (TPC) regarding their buffering capacity and their efficiency in recovering salmonella from feed samples.

For this, a total of 269 samples were cultured in both pre-enrichment broths, and initial and final pH values were determined. Interestingly, pH values of the TPC samples remained nearly neutral, and the pH values of the BPW samples were lower but the recovery of salmonella isolates was better in BPW samples.

Obviously, the choice of non-selective pre-enrichment broths should not only be dependent on the buffering capacity but also on other factors such as high versus low microbial loads in the feed, and the moisture content of the samples, etc.

The study is well done, and all necessary information is given. The introduction contains sufficient information, materials and methods are comprehensible. The results are presented in 3 tables, which provide a good and informative overview.

An improvement of the significance would have been possible with spiking experiments, which the authors unfortunately did not perform in the context of this study. The authors only state this in their conclusions as a topic for future studies (“Further research of feed inoculated with a known Salmonella strain and concentration and assayed with pre-enrichment TPC and BPW is necessary to determine their efficacy in recovering Salmonella”).

All in all, I recommend the acceptance of the manuscript. 

Reviewer 2 Report

Manuscript ID: animals-2579771

The subject of this work falls within the general scope of Animals Journal. This article aims to compare two pre-enrichment mediums, tris phosphate carbonate and buffered peptone water, for detecting Salmonella in feed ingredients.

The study is original and the results derived from it can be considered of interest for the scientific community. Furthermore, the manuscript is overall quite well written, and the conclusions are sound and consistent with the results.

However, I have some comments and suggestions for the author.

Specific comments:

Title

I suggest to respect delete the abbreviation (TPC) and (BPW) and respect the journal template

Introduction

Line 78: Replace project” withstudy”

Material and Methods

Please unified all the subsection title according to the journal template

The study design is well-written and allows reproducibility.

Results

Line 147, 171: correct pH writing

Discussion

Line 209-210: .. .. „DDGS and wheat middlings are by-product that vary in composition since the raw material used can be from different varieties, different environmental conditions, and the techniques to obtain the mentioned by-products can be different”..

Not only the chemical composition of these by-product varies, but the other feed ingredients composition (e.g., soybean meal, peanut meal or poultry and meat/bone by-products) also could be variable according to the varieties, climate conditions, different processing factors, etc. Can authors discuss more these?
